# Endozoochory by the cooperation between beetles and ants in the holoparasitic plant *Cynomorium songaricum* in the deserts of Northwest China

Zhi Wang[1,2☉], Huan Guan[1☉], Bingzhen Li[1], Qianqian Zhang[1], Qing Chen[3], Dehui Wang[1,4], Kexin He[1], Zikang Jin[1], Guilin Chen[1]*

1 Key Laboratory of Herbage & Endemic Crop Biology Ministry of Education, School of Life Sciences, The Good Agriculture Practice Engineering Technology Research Center of Chinese and Mongolian Medicine in Inner Mongolia, Inner Mongolia University, Hohhot, China, 2 Alxa League Forestry Grassland Research Institute, Alxa League, China, 3 Spallation Neutron Source Science Center, Institute of High Energy Physics, Chinese Academy of Science, Dongguan, China, 4 Department of Agriculture, Animal Husbandry, Forestry and Bioengineering, Xing An Vocational and Technical College, Xing'an League, China

☉ These authors contributed equally to this work.
* guilinchen61@163.com

## Abstract

*Cynomorium songaricum* Rupr. first described by Carl Johann (Ivanovič) Ruprecht in 1840 is a desert parasitic plant that mainly parasitizes the roots of *Nitraria* L. (especially of *Nitraria tangutorum* Bobrov., *Nitraria sibirica* Pall.). During seed maturation, *C. songaricum* releases a distinct smell, and its seeds are round and dust-like. Previous studies indicated that most parasitic plants produce small seeds, which are primarily dispersed by the wind. Recent studies reveal the significant role of animals in the seed dispersal of parasitic plants. In this study, we combined (1) the direct observation of the seed dispersal of *C. songaricum*, and (2) the indoor breeding of beetles and ants to assess the viability of seeds, clarify the seed dispersal system, and explore the mechanisms by which the seeds attract dispersal agents. By a population study, we identified beetles (*Mantichorula semenowi* Reitter, 1888) and ants (*Messor desertora* He & Song, 2009) as the primary seed dispersal agents for the *C. songaricum*. These plants rely on the visits from these agents to transfer their seeds near the roots of the host plant, *Nitraria* L.. The release of a distinct volatile compound from *C. songaricum* seeds attracts *M. semenowi* and *M. desertora* to consume and/or transport the seeds. This study provides the first evidence of a multi-medium and inter-species seed dispersal system in the *C. songaricum*. This study elucidates the role of invertebrates in the seed dispersal of desert parasitic plants. We propose that the two seed dispersal agents play distinct roles in the sequential seed dispersal of *C. songaricum*, representing two key stages in the overall seed dispersal mechanism.

**Data availability statement:** All relevant data for this study can be found in the figshare database, DOI: https://doi.org/10.6084/m9.figshare.28234862.v2.

**Funding:** This study was supported by the National Natural Science Foundation of China (30660015 and 31260117) and Inner Mongolia Science & Technology Plan (CGZH2018127 and 2021GG0152). The funders had no role in study design, data collection and analysis, decision to publish, or preparation of the manuscript.

**Competing interests:** The authors have declared that no competing interests exist.

## Introduction

Photosynthesis is generally considered to be a key characteristic of the non-parasitic plant. Therefore, the evolution of parasitic plants, which lack photosynthetic capability, remains one of the most interesting and challenging topics in plant biology [1]. Parasitic plants have evolved independently at least 12 times in angiosperms, and they can be found in almost every ecosystem [1–3]. These parasitic plants are classified into hemiparasites and holoparasites based on their ability to photosynthesize and their dependence on host plants for nutrients [1]. Hemiparasites, such as *Viscum* L. (mistletoe) and *Striga* Lour. (witchweed), possess chlorophyll and are capable of limited photosynthesis, yet they still depend on their host plants for water and nutrients. In contrast, holoparasites, including *Orobanche* L. (broomrape) and *Cuscuta* L. (dodder), lack chlorophyll entirely and rely exclusively on their host plants for both carbon and mineral nutrients. Hemiparasite plants have green leaves and can photosynthesize, while holoparasite plants cannot photosynthesize [4]. During the evolution of parasitic plants, many species have developed specific traits through the co-evolution with their host plants, such as a significant reduction in seed size and structural complexity [2,3,5]. The seeds of most parasitic plants, referred to as "dust seeds", are dispersed in the air like dust. A common characteristic of these dust seeds is the presence of cavities or holes [3]. In seed dispersal studies, dust and small seeds are typically considered to be wind-dispersal [5–7]. However, some studies have shown that the effectiveness of wind dispersal in parasitic plants may be affected by other factors such as the growth environment and seeds or fruit type. Other potential seed dispersal media, aside from wind, may also play a role in the transmission of parasitic plant seeds [8]. Previous research has indicated that, due to their extremely small size, parasitic plant seeds, may be dispersed endozoically by animals [3].

Currently, dust seeds are commonly found in holoparasite species across 10 angiosperm families, including Apodanthaceae, Balanophoraceae, Cynomoriaceae, Cytinaceae, Hydnoraceae, Lennoaceae, Mitrastemonaceae, Mystropetalaceae, Orobanchaceae, and Rafflesiaceae [9]. Reports on seed dispersal have been limited to the Balanophoraceae, Cytinaceae, and Orobanchaceae family. For instance, research indicates that the fruiting structures of *Balanophora yakushimensis* have evolved to facilitate seed dispersal by birds. The vibrant red color, lack of noticeable scent, and timing of winter fruiting make the infructescences particularly appealing to birds such as the red-flanked blue tail and pale thrush [10]. Suetsugu (2018a, 2018b) also discovered that camel crickets play a crucial role in the seed dispersal of *Phacellanthus tubiflorus* and *Yoania japonica* [11,12]. Additionally, beetles have been shown to disperse seeds of the parasitic plant *Cytinus hypocistis* [13]. To date, it has been recognized that during the seed dispersal process of parasitic plants with dust seeds, other potential seed dispersal vectors, aside from wind, are influenced by the environment of the parasitic plant habitat [3,8].

*Cynomorium songaricum*, which belongs to the genus *Cynomorium* from the Cynomoriaceae family, is a vital parasitic plant classified as endangered [14]. This species is well-known for its medicinal and nutritional benefits and is widely distributed across desert areas in Asia, Africa, and Europe [15]. In China, it is commonly used in traditional medicine for the liver and kidney, to replenish essence and blood, and to promote bowel movements [16]. Recent pharmacological studies have identified active components like phenolic acids and flavonoids in *C. songaricum* [17]. These components are known to possess antioxidant and antiviral properties, as well as to aid in the prevention and treatment of obesity, diabetes, Alzheimer's disease, and enhancing memory function [18,19]. Moreover, its nutritious content and appealing flavor have resulted in the development of various food products including *C. songaricum* wine, tea, and desserts [20], positioning it as a plant of significant culinary and medicinal research interest. However, limitations in availability and inconsistent quality have hindered the large-scale development and sustainable use of *C. songaricum* [15,21].

Previous research has demonstrated that *C. songaricum* attracts insects for pollination by releasing malodorous volatiles [22]. However, there are currently no reports on the seed dispersal biology of Cynomoriaceae. The *C. songaricum* population is a wild population that depends on *Nitrariaceae* as its host plant. The C. songaricum seeds are in contact with or closely situated on the ground, suggesting that they could be consumed by ground-dwelling animals. We investigated the seed dispersal system and the functional properties of *C. songaricum* seeds. The seeds share similar traits (dusty seeds, hard seed coat) with those of other holoparasites plants, which are dispersed by invertebrates. Therefore, we hypothesize that the seeds can pass through the digestive tract of animals. Additionally, the seed bears a subsidiary structure, described as an "elaiosome". When the seeds are brown and dried, the dry pulp appears to function as an elaiosome, facilitating removal by ants, we also hypothesize that ants play a role in the seed dispersal of *C. songaricum*.

Here, we aim to study the seed dispersal system of *C. songaricum*, including whether and which animals are involved in effective seed dispersal. Firstly, we conducted a field study of the visiting animals of *C. songaricum* plants to analyze the main kind of seed dispersal agents. Secondly, we identified the seed dispersal system using feeding experiments. This analysis allowed us to quantify the impact of living behaviors of dispersal agents on the dispersal mechanism of *C. songaricum* seeds, analyze the effectiveness of seed dispersal, and identify the source of seed dispersal agents. Thirdly, we conducted a seed trait analysis to study the unique micro-structure of *C. songaricum* seeds that are adapted for seed dispersal. Finally, we will discuss the evolutionary implications of seed dispersal strategies by dispersal agents on the desert ecological system.

## Materials and methods

### Study species and field sites

In this research, we conducted a field experiment during the fruiting period of the *C. songaricum* plants from June to August 2021 and 2022. The research was conducted in Alxa League, Inner Mongolia, China (N39°36', E105°05'). According to the Köppen - Geiger climate classification system, the research area is classified as a temperate semi-arid steppe climate (BSk), with an average annual temperature of 7.9-10°C and an annual precipitation of 39.3-224.2 mm [23,24]. We selected three areas for field observation of *C. songaricum* (S1 Table). Study site 1: The research base in Jilan Tai Town, Alxa Left Banner, located in the southeastern Tengger Desert (N39°36', E105°05'), with an elevation of 1200-1400 m. This site is characterized by arid conditions and a diverse plant community, with *C. songaricum* occurring naturally and parasitizing *Nitraria* L. as its host; Study Site 2: The research base in Yingen Sumu, Alxa Left Banner, located in the northeastern Tengger Desert (N38°27', E105°23'), with an elevation of 1200-1400 m. Similar to site 1, this region features semi-arid conditions and supports wild populations of *C. songaricum*, which parasitize *Nitraria* L.; Study Site 3: The research base in Ejina Banner, Alxa League, situated in the northwestern Badain Jaran Desert, with an elevation ranging from 1200 to 1700 m. This site is characterized by harsh desert conditions and sparse vegetation, with *C. songaricum* growing as a wild species and parasitizing *Nitraria* L.. The study site consisted mainly of shrub forests dominated by *Nitraria* L. (Nitrariaceae) (mainly of *Nitraria tangutorum* Bobrov; *Nitraria sibirica* Pall.). In this research, these research sites were selected because they are currently important protected areas for wild *C. songaricum* resources in China.

From June to August 2021 and 2022, as long as the weather permitted, two observers would conduct investigations on the insect visitation situation at the *C. songaricum* observation points of each research site for two to three days almost every month. A total of six rounds of

collection work were carried out. Observations were made once a day from sunrise to sunset. The first research site was selected according to a rotating schedule, and then the sites that had not been sampled were sampled in turn. The investigations at the study sites were only carried out on days when it was warm (air temperature ≥ 28 ± 3°C), sunny, and the wind speed was lower than 20 kilometers per hour. This was done to increase the probability of the activities of seed dispersers.

We used the anemometer of the portable weather station (FRT FWS02) in Alxa League, China to measure the wind speed. The sampling of seed dispersers was carried out between sunrise and sunset. Due to the large geographical span of each research site, the specific observation times were as follows: for study Site 1, it was between 6:00 am at sunrise and 7:30 pm at sunset; for study Site 2, it was between 5:30 am at sunrise and 9:00 pm at sunset; for study Site 3, it was between 5:30 am at sunrise and 9:30 pm at sunset. The nighttime investigations were only conducted when it was warm (air temperature ≥ 10 ± 3°C) by using remote cameras equipped with infrared motion sensors. All potential biological visitors were observed during all time periods [25,26].

During the period from June to August 2021 and 2022, each of the two observers observed the types, numbers, visit durations, and behavioral manifestations of potential seed visitors by patrolling around the research sites and sitting beside the *C. songaricum* populations. Each observation of each *C. songaricum* plant cluster lasted for 10 to 30 minutes, and the cumulative observation time reached 636 h (S2 Table).

All insects that obviously came into contact with the seeds of *C. songaricum* were recorded as visitors. The collected visitors were identified and classified to the morphological species level. All species identification work was completed with the help of dissecting microscopes, taxonomic literature, the assistance of taxonomic experts, and by comparing with reference specimens.

Investigations indicated that seed consumption mainly occurred during the day (Fig 1). Only species that participated in consuming and/or transporting the seeds were identified

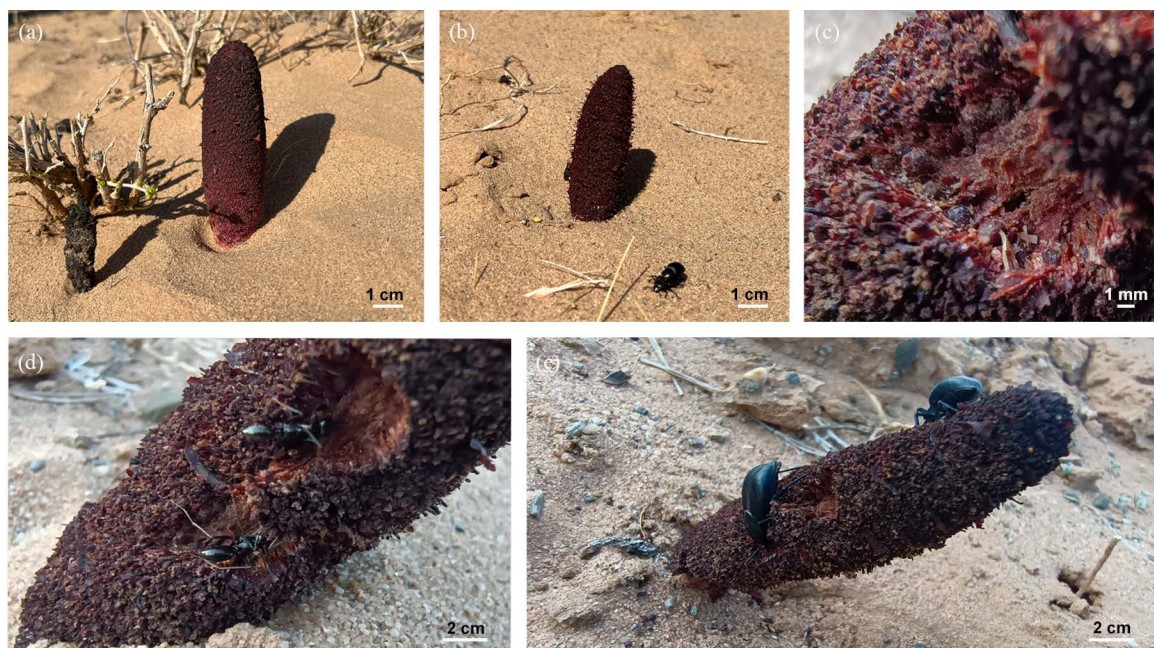

**Fig 1. The *C. songaricum* plant studied and the visiting insects.** (a) *C. songaricum* during the emergence period; (b) *C. songaricum* during the flowering period; (c) *C. songaricum* during the fruiting period; (d) Ants on *C. songaricum*, carrying seeds; (e) Beetles on *C. songaricum*, feeding on the seeds and fleshy stems. (a), (b) bars = 1 cm; (c) bar = 1 mm; (d), (e) bars = 2 cm.

as seed dispersal agents. This study involved the collection of plant and invertebrate samples from the wild *C. songaricum* resource protection area in Alxa League. This study has obtained the approval of both parties and does not require a field permit number.

## Field experiments

In this study, field observations were conducted at three sites between June and August in 2021 and 2022. Each site was observed for two-three days per month. In both 2021 and 2022, nine visits were made each year, with three visits made to each site respectively. Field observations revealed that throughout the study period, only beetles (*Mantichorula semenowi* Reitter, 1888) and ants (*Messor desertora* He & Song, 2009) would visit *C. songaricum* seeds at the three research sites. Therefore subsequent observations and experiments focused on *M. semenowi* and *M. desertora* as the primary seed dispersers. At each study site, interactions between *C. songaricum* and the two studied insects were assessed based on their activity duration and behaviors.

To assess the impact of *M. semenowi* and *M. desertora* on the seeds of *C. songaricum*, insect sampling was conducted at three research sites from July 4th to 11th, 2021, during the fruiting stage of the plant. Two sampling methods were used: direct capture for *M. semenowi* and Kule-PRO aspirator for *M. desertora*. The Kule-PRO aspirator operates by activating the blower motor, which creates airflow through the pipe to attract ants into a 50 mL centrifugal pipe. For each study site, 50 *M. semenowi* individuals were captured from the *C. songaricum* populations, with a total of 150 *M. semenowi*. Each *M. semenowi* was placed in a separate 3 × 3 × 4 cm plastic container and incubated in the shade for 10 to 12 h. During this period, 347 fecal samples were collected, with only complete *C. songaricum* seeds detected in the feces of 71 *M. semenowi*. The number of fecal samples containing complete seeds was quantified. Statistical analysis revealed that 178 fecal samples contained a total of 1,045 *C. songaricum* seeds.

Meanwhile, we conducted insect sampling using a Kule-PRO aspirator, capturing 150 *M. desertora*, the same number as the *M. semenowi*. During the entire observation period, we employed random sampling for data collection. Specifically, for three consecutive days, 10 *M. desertora* and 10 *M. semenowi* were randomly selected each day from those visiting the flowers. Once an *M. desertora* began to transport seeds, its movement was tracked until it entered the nest, which was recorded as valid seed-carrying behavior. Once an *M. semenowi* began to feed on the seeds, it was tracked until the seeds were consumed, which was recorded as valid gnawing behavior. Considering the overall visitation frequency and duration, we calculated the average time *M. desertora* spent carrying seeds back to the nest and the time *M. semenowi* spent gnawing on the seeds, along with the number of seeds transported.

The field experiment aimed to investigate the interaction between the main visitors of *C. songaricum-M. desertora* and *M. semenowi-*and its seeds, to determine whether these insects act as seed dispersers or seed predators.

## Feeding experiments in the laboratory

The feeding experiment was conducted in September 2022 and lasted for 21 days. Both *M. semenowi* and *M. desertora* were subjected to feeding experiments in a laboratory setting under controlled conditions. Specifically, for *M. semenowi*, the breeding chamber was maintained at a constant temperature of 25 ± 2°C, with relative humidity ranging from 40% to 60%. For *M. desertora*, the artificial nests were kept under the same temperature conditions, with humidity precisely controlled between 50% and 60%. During the experiment, both species had unrestricted access to water, and *C. songaricum* seeds were the sole food source.

Sixty *M. semenowi* were randomly selected from the 150 *M. semenowi* captured in the wild, with 20 *M. semenowi* from each research site. The 60 *M. semenowi* were then divided into two groups, with 30 *M. semenowi* in each group, consisting of 10 *M. semenowi* from each site. All *M. semenowi* were fasted for 48 hours before the start of the rearing experiment. Subsequently, individual and mixed-rearing feeding experiments were conducted [27,28]. The specific experimental procedures were as follows:

Individual rearing: Thirty *M. semenowi* captured in the wild were placed in 30 separate rearing rooms (The rearing room is a transparent cylinder, 8 cm in bottom diameter and 10 cm high, with a grid-covered top for gas exchange and observation).

Mixed rearing: Thirty *M. semenowi* captured from the wild were placed in three independent rearing rooms, with 10 *M. semenowi* in each room (The rearing room is a transparent cuboid, measuring 20 cm in length, 10 cm in width and 15 cm in height. Its top is fitted with a grid cover, facilitating gas exchange and observation).

A complete feeding experiment involved collecting *M. semenowi* feces from each rearing room at 12, 24, and 48 h after the first feeding, and observing and recording the number of intact *C. songaricum* seeds in the feces. Three rounds of feeding experiments were conducted.

In the individual rearing, 10 *C. songaricum* seeds were provided to each *M. semenowi* in one feeding experiment. For 30 *M. semenowi*, 300 seeds were provided. With six times repetitions, a total of 1800 seeds were used.

In the mixed rearing, the same number of 10 *C. songaricum* seeds were provided to each *M. semenowi* in one feeding experiment. Each rearing box contained 10 *M. semenowi*, with 100 *C. songaricum* seeds provided per box, totaling 300 seeds. With six times repetitions, a total of 1800 seeds were used.

In the feeding experiment, each *M. semenowi* underwent three duration treatments: 12, 24, and 48 h. At each time point, *M. semenowi* feces were collected from rearing boxes to count the number of fecal samples and intact *C. songaricum* seeds. An independent-sample t-test analyzed differences in seed excretion in its feces under different rearing modes.

Additionally, in the ant-rearing experiment, ants were captured from three study sites. Each group of 50 *M. desertora* was placed in a separate colony, resulting in three colonies with a total of 150 *M. desertora*. However, during the study, we observed that the *M. desertora* did not carry the seeds of *C. songaricum*, which contradicted our field observations. This may be attributed to the absence of a queen ant in the colony [29].

We purchased three sets of ant colonies of the target species *M. desertora* from a biotechnology company. Each colony consisted of one queen ant, ten soldier ants, and 120 worker ants. Immediately thereafter, we initiated the rearing experiment. The specifications of the artificial ant nests were as follows: the nest body measured 205 × 125 × 40 mm, constructed from mixed materials to provide adequate living space; the activity area measured 205 × 125 × 110 mm, allowing the ants to carry out their daily activities; and two independent water towers, each with a capacity of 18 mL, were included to maintain moisture levels and satisfy the ants' drinking water and humidity requirements. During the experiment, each ant colony was provided with 100 *C. songaricum* seeds, amounting to a total of 300 seeds across the six times repetitions. A total of 1800 seeds were used. The seeds were collected from the ant nests at 6, 12, 24, and 48 h, and the number of intact seeds was counted. Throughout the experiment, only *C. songaricum* seeds and necessary water were provided to minimize external interference. In the above-mentioned feeding experiments, before each new round of repetition is initiated, the experimental subjects will be subjected to a 48-hour starvation treatment.

## Dissection experiment

The dissection experiment was conducted in November 2022 and lasted for 7 days. Start a new feeding experiment. This experiment randomly selected three *M. semenowi* at four-time points (6, 12, 24, and 48 h after the feeding experiment) for dissection, and no replenishment of *M. semenowi* was made, so it was not included in the rearing experiment. At each time point, a total of six *M. semenowi* were used, with three from the individual-rearing experiment and three from the mixed-rearing experiment. The experiment procedure was as follows:

After each sampling, the *M. semenowi* were immersed in 75% ethanol for anesthesia. Then, the anesthetized *M. semenowi* was fixed onto a wax plate. An anatomical scalpel was used to cut open the intersegmental membrane of the *M. semenowi* abdomen. An anatomical needle was used to separate the connective tissue around the intestine, and the intestine was fully excised and placed in a culture dish containing physiological saline. Then, the intestine was examined under a stereomicroscope for the presence of *C. songaricum* seeds. Some of the *C. songaricum* seeds obtained from the intestine were used for scanning electron microscope observation, while others were used for the subsequent seed viability test.

## Seed viability

The seed viability test was conducted in November 2022 and lasted for 10 days. Mechanical damage to seeds caused by chewing is a common phenomenon in many plant species. To assess the impact of *M. semenowi* and *M. desertora* on *C. songaricum* seeds, we conducted seed viability tests under different treatments: Seeds directly collected from *C. songaricum*; Seeds with only the elaiosome removed by *M. semenowi* and *M. desertora*; Seeds collected from the intestines of *M. semenowi* at four retention times (6, 12, 24, and 48 h); Seeds collected from *M. desertora* nests at four retention times (6, 12, 24, and 48 h). The experiment procedure was as follows: We then used 2, 3, 5-triphenyltetrazolium chloride (TTC) to stain. The TTC test differentiates viable seeds from non-viable ones based on their respiration in a hydrated state. Living cells will change from colorless to pink/red. Furthermore, we applied a combination of techniques described by De Vega and De Oliveira that allowed us to observe whether seeds retained normal embryos after being consumed by animals [30].

## Attract seed dispersers

Field observation revealed that the above-ground parts of *C. songaricum* have a brown color, which contrasts sharply with the surrounding environment (Fig 1a). Additionally, the seeds often emit a greasy odor that is perceptible to humans (Wang, field observation). This study aimed to examine visitors' responses to both visual (fruit color) and olfactory (odor) cues at three research sites. Two treatment groups were set up in the experiment, each repeated for 7 consecutive days. The number and types of visitors were recorded daily from sunrise to sunset. Specifically, we tested the visitors' response to the fruit sequence (view signals) and seed scent (odor signals). Two treatment groups were set up in the wild. **Treatment A** involved covering the infructescence with yellow gauze mesh bags to isolate the visual signal (color) of *C. songaricum*, while the control group was covered only with yellow mesh bags to eliminate material interference (to test the visitors' response to *C. songaricum* odor signals); **Treatment B** involved the infructescence was covered with a transparent plastic bag to block the scent of *C. songaricum* seeds, while the control group was only covered with plastic bags to eliminate material effects (to test the visitors' response to *C. songaricum* view signals); The control group involved unbagged *C. songaricum*, the distance between each treatment was approximately 10 m. The experiment ran continuously for 7 days, with visitors the types and numbers recorded from 6:00 am to 8:00 pm.

## Statistical analysis

A direct comparison was made of the visiting time and frequency of each visitor (*M. semenowi* and *M. desertora*) to *C. songaricum*. We used Origin 2022 to construct graphs and tables displaying the visiting frequency and duration to evaluate the proportional contributions of *M. Semenowi* and *M. desertora* among the visitors to *C. songaricum*. To explore the impact of the studied species on *C. songaricum* seeds, we conducted a statistical analysis using SPSS 26 software. Prior to further analysis, the Shapiro-Wilk test was applied to assess the normality of the data and evaluate the homogeneity of variance (ANOVA). The following data were analyzed: 1) Seeds collected by *M. semenowi* at three-time points under different rearing modes; 2) Seeds collected from *M. desertora* nests at three-time points. The mean and standard deviation of the number of seeds consumed by *M. semenowi* and *M. desertora* at each time point were calculated. To determine whether *C. songaricum* attracts visitors via visual or olfactory signals, we used the number and frequency of visitors during different time periods in the field experiment to generate bar charts using Origin 2022.

## Results

### Seed visitor observation

Detailed time-dependent observations were conducted to monitor the different species of insects that visit the plant (S1 Fig). Direct observation revealed that invertebrates, *M. semenowi* and *M. desertora* are the major visitors of the seeds of *C. songaricum* (Table 1). The feeding duration of *M. semenowi* and *M. desertora* within a *C. songaricum* cluster was found to be 10 - 12 hours per day (16 ~ 36°C). We captured 150 *M. semenowi* and collected 347 fecal samples. Of these, 178 fecal samples contained intact *C. songaricum* seeds, accounting for 51.29%. In addition, *M. desertora* frequently visited the plant and carried seeds back to their nests. At the three study sites, we precisely measured that it took *M. desertora* 14.25 ± 5.91 minutes (mean ± SD) to gnaw a *C. songaricum* seed and transport it to their nest (S3 Table). On average, *M. desertora* transported 144.27 ± 29.94 (mean ± SD) *C. songaricum* seeds into their nests (S4 Table). No vertebrates were observed to visit or consume the seeds of *C. songaricum*. *M. semenowi* were identified as the primary invertebrates that consume the majority of seeds. Additionally, camera recordings equipped with motion sensors confirmed that other species no other species visited the *C. songaricum* seeds at night (air temperature ≥ 10 ± 3°C).

### *M. semenowi* foraging activity

*M. semenowi* foraging activity is mainly concentrated in the morning (6:30-11:30) and evening (15:30-19:00) (Ambient temperature: 18°C ~ 33°C, ± 2°C). At noon (12:00-15:30), when the ground temperature exceeds 50°C, *M. semenowi* activity decreases and they bury themselves in the sand, disappearing from view. The foraging activity times of *M. semenowi* across different research sites were generally consistent with the patterns described above. Although some differences were observed, they were not statistically significant (S1 Fig). All *M. semenowi* collected while feeding on *C. songaricum* seeds in the wild excreted intact *C. songaricum* seeds.

**Table 1. Identity of *C. songaricum* seed consumers and the duration of their visits within a 24-hour period.**

| Animal taxa | Day-time (h/ h) | | |
|---|---|---|---|
| | Ejina*(n) | Jilantai*(n) | Yingen*(n) |
| *Mantichorula semenowi* Reitter, 1888 | (10.59 ± 0.40)/ 24 | (10.82 ± 0.44)/ 24 | (10.21 ± 0.35)/ 24 |
| *Messor desertora* He & Song, 2009 | (11.81 ± 0.29)/ 24 | (11.73 ± 0.29)/ 24 | (11.44 ± 0.30)/ 24 |

Microscopic observation revealed that the seeds recovered from the feces of *M. semenowi* remained intact, with no deformation of the embryos (S2 Fig).

The *M. semenowi* feeds on mature brown seeds with dried fruit color and is present concurrently with ants on the infructescence. The *M. semenowi* feeds on infructescences alone until the dried flesh of the embedded seeds is consumed. Observations of *M. semenowi* revealed the average time for *M. semenowi* to consume a seed in the field was 13.8 ± 2.8 s (mean ± SD, n = 9) (S5 Table). At noon, as the temperature gradually increased, this environmental change caused the *M. semenowi* to leave the *C. songaricum* infructescence and return to their nests. Our observations revealed that the *M. semenowi* constructed their nests up to 50 centimeters away from the feeding site, which is the *C. songaricum* infructescence. *M. semenowi* forage near the *C. songaricum* infructescence. This behavior ensures that more seeds remain in the vicinity of the mother plant, thereby increasing the likelihood of seed deposition around the plant.

We conducted a detailed analysis of the intact *C. songaricum* seeds present in *M. semenowi* feces collected at different time points under varying rearing modes. The results revealed no significant difference in the number of intact *C. songaricum* seeds collected after each round of rearing experiments between the two rearing modes. Specifically, the number of *C. songaricum* seeds collected under the individual rearing mode was 286 ± 3, while the number collected under the mixed rearing mode was 289 ± 6 (Table 2).

Further analysis revealed that regardless of the rearing mode (individual or mixed), *M. semenowi* excreted a significant number of seeds within 12 hours after feeding. Specifically, under the individual rearing mode, 65.04% of the seeds were excreted, totaling 186 ± 3 (n = 286), whereas under the mixed rearing mode, 57.79% of the seeds were excreted, totaling 167 ± 5 (n = 289, Table 2). Following one round of rearing experiments, we counted the number of seeds that passed through the *M. semenowi* intestines. In the individual rearing mode, 240 ± 9.46 seeds passed through the intestines, with approximately 80% excreted, while in the mixed-rearing mode, 231 ± 11 seeds passed through the intestines, with approximately 76.9% excreted. These results indicated that more than 50% of the *C. songaricum* seeds were excreted within 12 h after feeding, and over 75% were excreted within 48 h (Table 2). Overall, there was no significant difference in the total number of intact *C. songaricum* seeds excreted by *M. semenowi* under the two rearing modes. However, certain differences were observed in specific time periods, possibly due to the competition effect. In the mixed rearing mode, *M. semenowi* may have hoarded more seeds in their bodies, leading to changes in the number of seeds excreted during some individual time periods.

## Observation of *M. semenowi* intestine

Additionally, we collected fecal samples at 6 h, 12 h, 24 h, and 48 h after feeding the *M. semenowi*, and screened *C. songaricum* seeds from these fecal samples under a microscope (Fig 2).

Table 2. Intact *C. songaricum* seeds in *M. semenowi* feces under different feeding modes.

| Seed type | Feeding methods | |
|---|---|---|
| | Individually feeding (mean ± SD) | Mixed feeding (mean ± SD) |
| Ungnawed seeds | 16 ± 2 | 26 ± 3 |
| Gnaw seeds | 30 ± 4 | 33 ± 5 |
| Seed-12 h | 186 ± 3 | 167 ± 5 |
| Seed-24 h | 36 ± 3 | 61 ± 4 |
| Seed-48 h | 19 ± 4 | 3 ± 2 |
| Total | 286 ± 3 | 289 ± 6 |

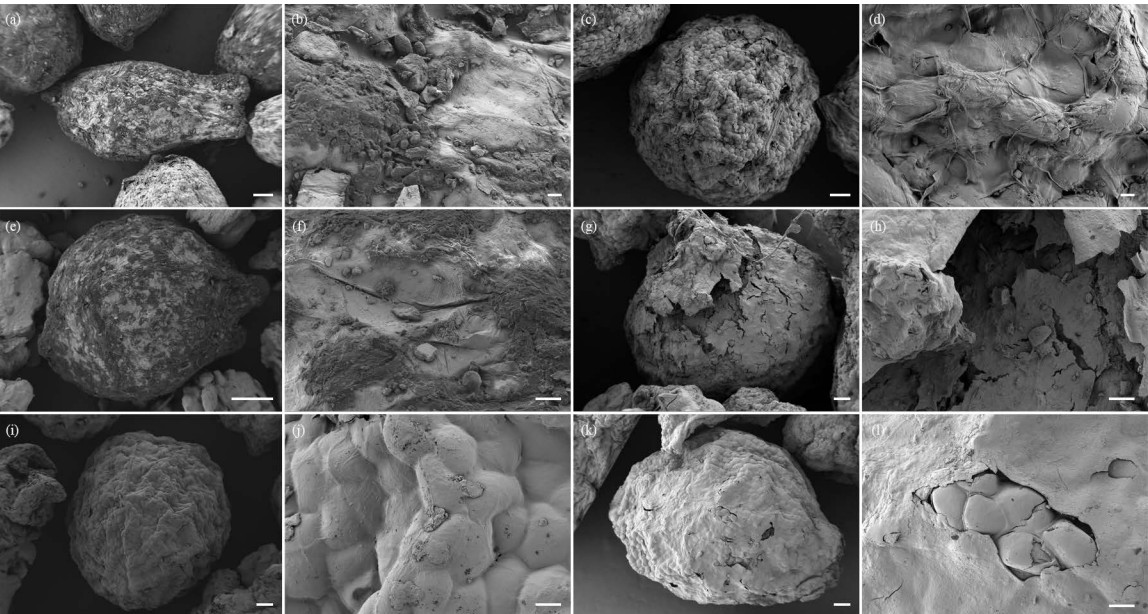

**Fig 2. Scanning electron microscopy of intact *C. songaricum* seeds collected from *M. semenowi* feces at different times.** (a-b) natural seeds. (c-d) non-elaiosome seeds. (e-f) intact seeds collected from *M. semenowi* feces 6 h later. (g-h) intact seeds collected from *M. semenowi* feces 12 h later. (i-j) intact seeds collected from *M. semenowi* feces 24 h later. (k-l) intact seeds collected from *M. semenowi* feces 48 h later. **(a)**, **(c)**, **(e)**, **(g)**, **(i)**, **(k)**, bars = 100 μm, **(b)**, **(d)**, **(f)**, **(h)**, **(j)**, **(l)**, bars = 10 μm.

Using scanning electron microscopy, we found that there was no difference between the seeds in feces after the *M. semenowi* had eaten for 6 h and the seeds in the natural state (Fig 2e-2f). Only the outer styles of *C. songaricum* seeds are consumed, and the damage to the seeds is minimal; 12 h after the *M. semenowi* eats, seeds are found in the feces, with the outer styles consumed and damaged, and part of the seed coat is exposed (Fig 2g-2h); 24 hours after feeding, the seeds found in feces showed that external style had been completely consumed, with the entire seed coat exposed and the surface smooth (Fig 2i-2j); 48 h after the *M. semenowi* had eaten, the seeds found in the feces the external style were completely consumed, with the entire seed coat exposed (Fig 2k-2l). The *C. songaricum* seeds were slightly damaged (mean = 1.83 ± 0.8, n = 300) (S6 Table).

Dissections of the digestive tract at different time intervals (6 h, 12 h, 24 h, and 48 h) after *M. semenowi* feeding further confirmed our results, demonstrating that *C. songaricum* seeds can evade the digestive action of the *M. semenowi* (Fig 3 and S7 Table). Intestinal dissection of the *M. semenowi* for 6 h found that a large number of *C. songaricum* seeds accumulated in the *M. semenowi* crop, with some of the seeds had entered the end of the intestine (mean = 23.67 ± 1.53, n = 30) (Fig 3a-3c); Intestinal dissection of *M. semenowi* after 12 h revealed that only a few *C. songaricum* seeds were accumulated in the *M. semenowi* crop, with no seeds found in the intestines (mean = 3.3 ± 0.58, n = 30) (Fig 3d-3f); Intestinal dissection of *M. semenowi* after 24 h revealed that *C. songaricum* seeds no accumulation of in the *M. semenowi* crop, and only 1 to 3 *C. songaricum* seeds were found in the intestines (mean = 1.67 ± 0.58, n = 30) (Fig 3g-3i). Intestinal dissection of the *M. semenowi* after 48 h showed that *C. songaricum* seeds did not accumulate in the *M. semenowi* crop. *C. songaricum* seeds have not been found in the intestines (mean = 0 ± 0, n = 30) (Fig 3j-3l).

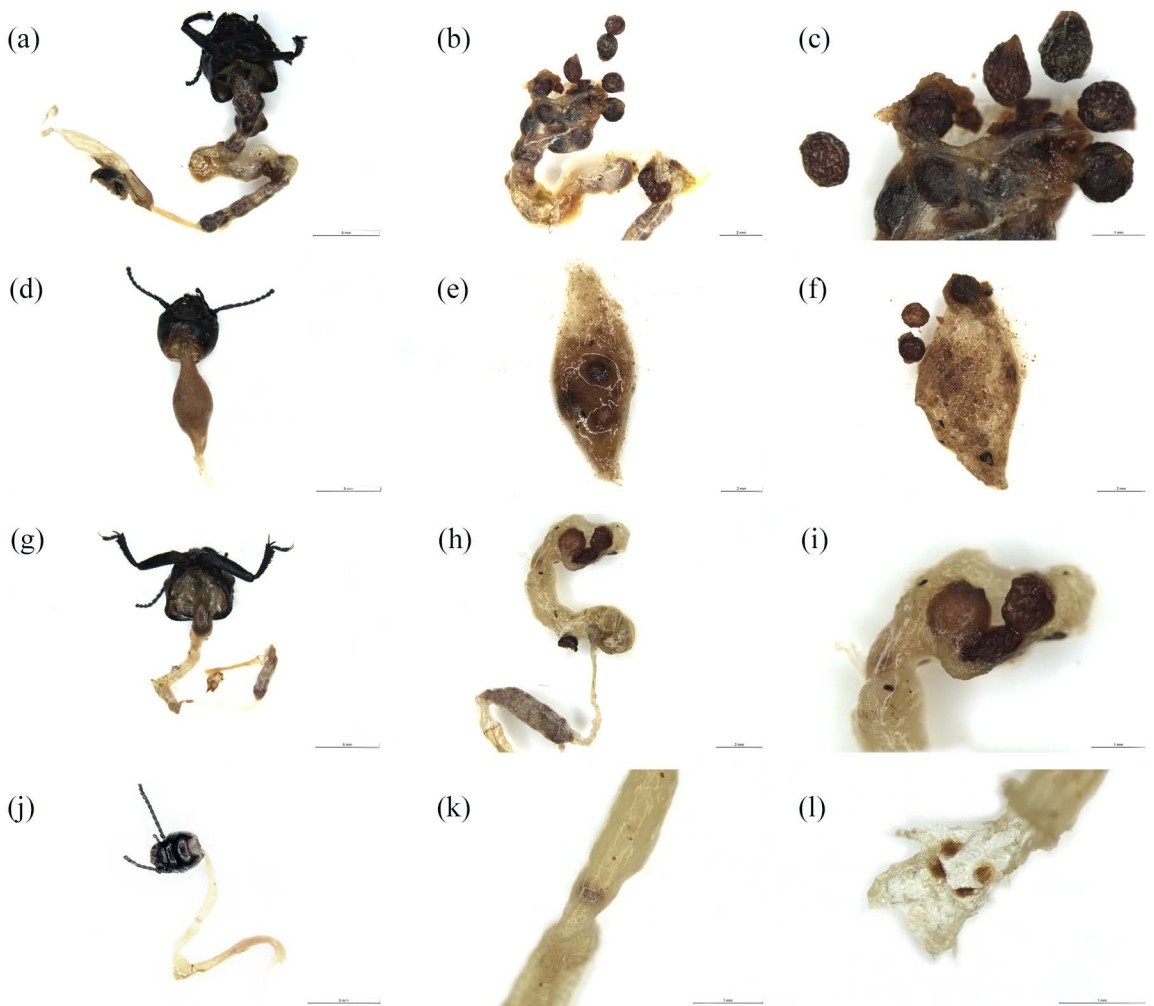

**Fig 3. Anatomy of the digestive tract of *M. semenowi* after feeding at different times.** (a-**c**) Anatomy of the digestive tract of *M. semenowi* 6h after feeding; (d-**f**) Anatomy of the digestive tract of *M. semenowi* 12h after feeding; (g-**i**) Anatomy of the digestive tract of *M. semenowi* 24h after feeding; (j-**l**) Anatomy of the digestive tract of *M. semenowi* 48h after feeding; (**a**), (**d**), (**g**), (**j**), bars = 5mm; (**b**), (**e**), (**f**), (**h**), bars = 2mm; (**c**), (**i**), (**k**), (**l**), bars = 1mm.

### *M. desertora* foraging activity

*M. desertora* foraging is primarily concentrated in the morning (6:30-12:30) and evening (15:00-19:30) (Ambient temperature: 16°C ~ 36°C, ± 3°C). At noon (12:30-15:00), when the ground temperature exceeds 55°C, *M. desertora* activity decreases sharply and the *M. desertora* retreat to their nests. When fruits are brown and dried, the dry pulp appears to function as an elaiosome and seeds are removed by *M. desertora* (S3 Fig). As the temperature rises at noon, the *M. desertora* seeks shelter to find shelter from the heat. *M. desertora* were observed building nests at distances ranging from 1 m to 10 m away from the feeding site. This behavior also causes the seeds to disperse from the vicinity of the parent plant. Although *M. desertora* movement was observed, the foraging activities of ants in our study system were spatially restricted to a small area. Additionally, since *M. desertora* transports seeds to their nests, it is difficult to detect the *C. songaricum* seeds carried by ants to the *M. desertora* nest in the wild. Therefore, we conducted observations by artificially simulating *M. desertora* nests. Using

stereomicroscopy, we observed that the seed coats of *C. songaricum* seeds in the *M. desertora* nests remained intact throughout the 6 - 48 h period. *M. desertora* primarily consumed the styles attached to the seeds. At 6 and 12 h, some seeds retained intact styles, while by 24 h, none of the seeds had complete appendages (Fig 4).

## Seed viability

The seed viability assessment showed that seeds recovered from *M. semenowi* droppings and seeds retrieved from *M. desertora* nests were still viable and showed damage to their embryos (S4 Fig). After staining with TTC, surviving embryos of *C. songaricum* appeared dark pink or red. According to TTC tests, no significant difference was observed in seed viability between natural *C. songaricum* seeds and *C. songaricum* seed samples from *M. semenowi* droppings and *M. desertora* nests (Table 3, S5 Fig).

## The mechanism of attraction of seed dispersal agents

Field experiments showed that when *C. songaricum* was completely exposed, the number of visits by *M. semenowi* and *M. desertora* was 257; in the odor treatment group, the number of visits by *M. semenowi* and *M. desertora* was 204; in the odor control group, the number of visits by *M. semenowi* and *M. desertora* was 112; in the visual treatment group, the number of visits by *M. semenowi* and *M. desertora* was 136; In the vision control group, the number

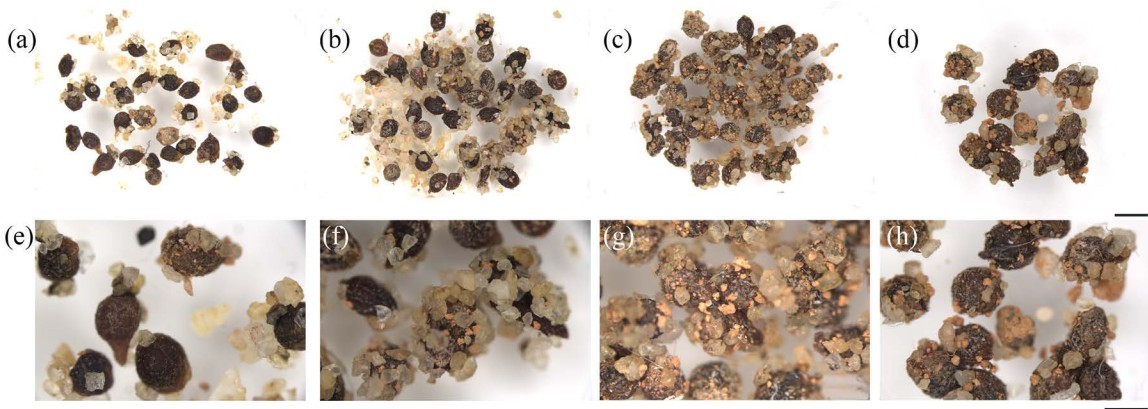

**Fig 4. The morphology of *C. songaricum* seeds after staying in the *M. desertora* nests after increasing time durations at a macroscale.** (a, **e**) The state of the seeds after staying in the *M. desertora* nest for 6 h; (b-f) the state of the seeds after staying in the *M. desertora* nest for 12 h; (c-g) the state of the seeds after staying in the *M. desertora* nest for 24 h; (d-h) the state of the seeds after staying in the *M. desertora* nest for 48 h.(a-**d**) Bars = 2 mm. (e-**h**) Bars = 1 mm.

**Table 3. Portion of the active seeds of *C. songaricum* in fecal and *M. desertora* nest.**

| Seed type | Species of consumers | |
|---|---|---|
| | *M. semenowi* (%) | *M. desertora* (%) |
| Ungnawed seeds | 83.64 ± 4.87 | 82.67 ± 0.82 |
| Gnaw seeds | 82.27 ± 2.02 | 81.50 ± 1.05 |
| Seed – 12 h | 81.86 ± 2.67 | 81.33 ± 1.21 |
| Seed – 24 h | 80.25 ± 1.39 | 80.17 ± 1.33 |
| Seed – 48 h | 77.38 ± 1.80 | 80.67 ± 0.82 |

of visits by *M. semenowi* and *M. desertora* was 87; when *C. songaricum* was fully exposed, the number of visits by *M. semenowi* and *M. desertora* was significantly higher than that the other groups ($p < 0.01$). The number of visits by *M. semenowi* and *M. desertora* in the treatment and visual treatment groups, as well as their corresponding control groups, was significantly different from the other groups ($p < 0.05$); the difference in the number of visits by *M. semenowi* and *M. desertora* between the odor control and visual control groups was not significant (Fig 5 and S8 Table). Therefore, it is suggested that when *C. songaricum* seeds are dispersed by animals, odor plays the primary role in attraction, while color serves a supporting role. The triangles represent the total visit frequencies of *M. semenowi* and *M. desertora* to *C. songaricum* in different time periods.

## Importance of *M. semenowi* and *M. desertora* to *C. songaricum*

In this study, we observed that *M. semenowi* and *M. desertora* visited each *C. songaricum* fruit cluster multiple times before the above-ground parts were depleted, during which the insects continuously visited, gnawed, and transported the seeds of *C. songaricum*. In addition, *M. semenowi* generally built their nests on the host plant, *Nitraria* L. colony, and we observed that more than ten *M. semenowi* concurrently nests on a single *Nitraria* L. colony, while ant nests could extend to depths of ($3 \pm 1$) meters underground [31]. Furthermore, this study found that the *Nitraria* L. colony has a well-developed underground root system, and the parasitic parts of *C. songaricum* and *Nitraria* L. colony are often located on the same two-dimensional plane at a depth of ($3 \pm 1$) meters. In this way, seeds spread by *M. semenowi* and *M. desertora* indirectly increase the likelihood of seed distribution near the roots of the host plant.

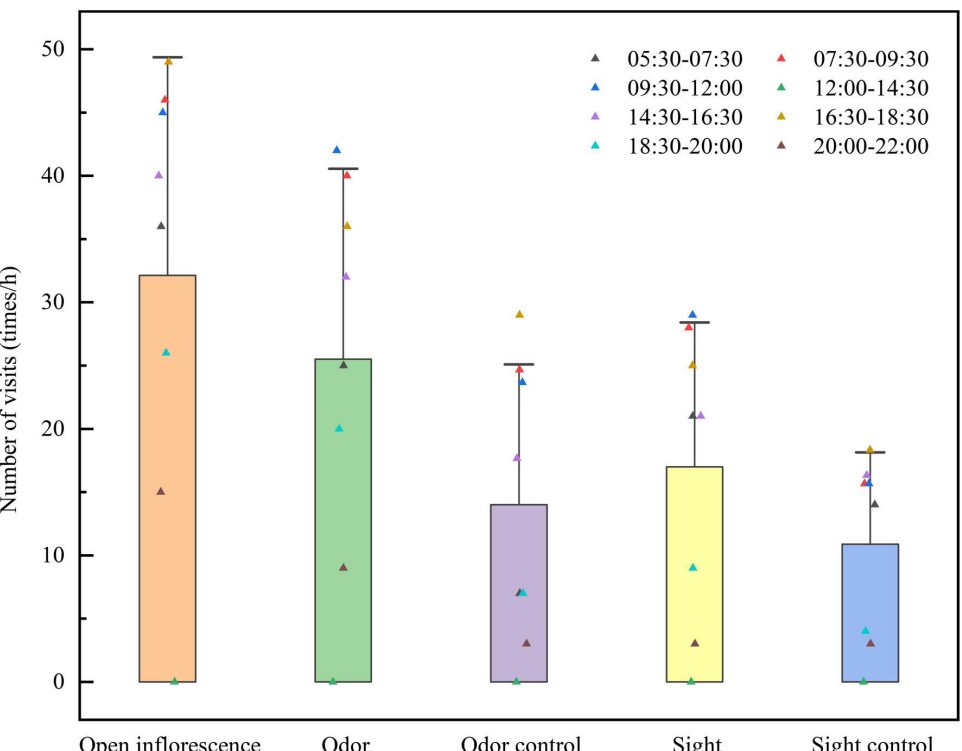

**Fig 5. Visiting frequency of insects to the *C. songaricum* infructescence under different treatments.**

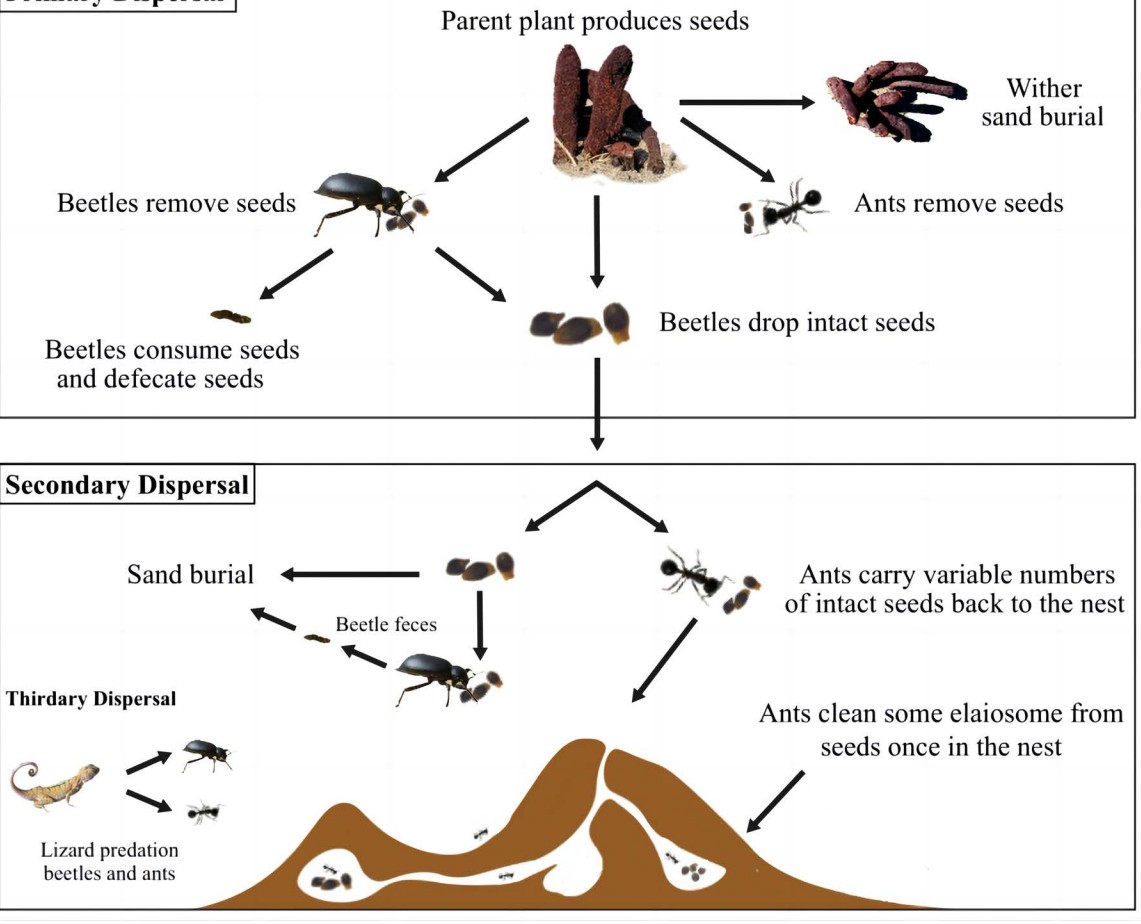

**Fig 6. Schematic diagram of *C. songaricum* seed propagation.**

From the above analysis, we can infer the seed dispersal mechanism of the *C. songaricum* plant (Fig 6): The first stage is the primary dispersal, which includes *M. semenowi* feeding, *M. desertora* transportation, and wind burial; the second stage is secondary dispersal, which involves seeds dropped during *M. semenowi* feeding in the first stage and seeds blown away by the wind. *M. semenowi* and *M. desertora* further consume and transport the seeds, and wind disperses them over long distances.

## Discussion

This study provides the first evidence that *M. semenowi* and *M. desertora* serve as seed dispersal agents for the *C. songaricum* plant species. *M. semenowi* and *M. desertora* play a crucial role as seed dispersers. Compared to most other parasitic plants, *C. songaricum* displays certain morphological features that can be considered adaptations for animal endozoochory [3,5,13]. The seeds of the *C. songaricum* plant are embedded in the fleshy stem (Fig 1c), which may facilitate *M. semenowi* ingesting the seeds while feeding on the plant, possibly serving as a reward for the insects. Additionally, the hard seed coat of *C. songaricum* seeds and the "elaiosome" (dried pulp) on the outer surface of the seeds protect them as they pass through the *M. semenowi* digestive tract [32–34]. Although most parasitic plant seeds may lack the strength

to withstand the chewing of ground *M. semenowi*, some evidence supports the hypothesis that beetles can also disperse parasitic plants [13].

It is commonly believed that in windy desert habitats, plants primarily rely on wind for seed dispersal [35–37]. However, wind-driven seed deposition driven by wind is unpredictable, and *C. songaricum* seeds may be dispersed randomly potentially far from the host plant. Relying solely on wind for seed dispersal may threaten the parasitic relationship between *C. songaricum* offspring and the host plant roots, hindering the reproduction of the *C. songaricum* population. Previous studies suggest that the effective seed dispersal of parasitic plants depends not only on the distance of seed dispersal but also on the proximity to suitable hosts [13]. Most parasitic plant seeds are small and require chemical signals from the host plants to successfully germinate, attach, and develop. *M. semenowi* play a crucial role in dispersing *C. songaricum* seeds, as they not only eat and consume the seeds but also gnaw on the fleshy stems, causing the seeds to detach. In our study, although damage to the seed coat of *C. songaricum* may negatively affect survival in the wild, we found that only a small proportion of seeds collected after 48 h were damaged. Feeding experiments also showed that more than 50% of *C. songaricum* seeds were expelled from the *M. semenowi* bodies 12 h after consumption, with over 75% eliminated 48 h post-feeding (Table 2). Therefore, we conclude that the beetles are seed dispersers of *C. songaricum*, rather than seed predators. We found that *M. desertora* can only carry seeds that have fallen after *M. semenowi* has gnawed on the fleshy stems of *C. songaricum*. *M. desertora* takes these seeds to the nests and then consumes the structures attached to the outside of the seeds. Therefore, *M. desertora* often carries seeds from areas where *M. semenowi* have previously fed. The seeds of *C. songaricum* are either consumed or dropped when transporting *M. semenowi* gnaw on the fleshy stems of *C. songaricum*. Furthermore, the habitats in which *C. songaricum* grows are not easily affected by wind and sand, with the seeds primarily being buried in the sand, which results in increased intraspecific competition. The seed dispersal behaviors of *M. semenowi* and *M. desertora* represent a mutually beneficial symbiosis. *M. semenowi* and *M. desertora* which are attracted by seed appendages can effectively transport *C. songaricum* seeds away from the parent plant, thereby reducing intraspecific competition [38].

Seed dispersal is typically a multi-stage process involving two or more distinct dispersal mechanisms. Regarding the multi-modal seed dispersal mechanism, the primary benefit of the first dispersal stage is typically predator evasion and the colonization of new species patches, while the primary benefit of the second dispersal stage is predator avoidance and directional diffusion [39–41]. Through directional dispersal, plants aim to transport their seeds to microsites conducive to the successful establishment of seedlings. As observed in our study, *M. semenowi* and *M. desertora* transport seeds to their nests through foraging behavior, which indirectly enhances the likelihood of contact with the roots of their host plants (*Nitraria* L.), thereby contributing to reproductive success [42]. To date, there is a lack of research examining the potential dispersal of *C. songaricum* seeds by invertebrates. Considering the duration and range of time that seeds remain in the digestive system, animals, particularly *M. semenowi*, may enhance the likelihood of seed dispersal in large quantities and over considerable distances from the plants. The multi-modal dispersal mechanism employed by *C. songaricum* may partially account for its unusually wide geographic distribution. The behaviors of different individual animal populations are crucial for the long-term survival of *C. songaricum*. The overlapping ecological niches between *C. songaricum* and its seed dispersal agents are crucial for the conservation of wild populations and the genetic diversity of this valuable species of *C. songaricum* species [43].

Although insects are seldom regarded as seed dispersal agents, it is important to note that growing evidence indicates invertebrates (e.g., crickets, beetles, ants, etc) play a crucial role in

the seed dispersal system of parasitic plants. In-depth studies have demonstrated that when seeds can be ingested as a whole and possess structural features to remain intact after passing through the digestive tract, endozoochory by animals becomes feasible [13,44]. Therefore, other seed-consuming animals are likely to contribute to seed dispersal, particularly for plants with small-sized seeds. Moreover, since *M. semenowi* and *M. desertora* are common members of ground-dwelling insect communities and have significant impacts on the desert ecosystems [45], they can serve as effective seed dispersal agents.

It is important to note that the seeds of *C. songaricum* emit a distinctive oily odor, which can be detected by both humans and *M. semenowi*. *M. desertora* are known to be attracted to the distinctive odor, acting as seed dispersers [46–49]. Given that the seeds of *C. songaricum* have a brownish-yellow coat, and are located close to the ground, *M. semenowi* and *M. desertora* primarily rely on olfactory signals to detect these seeds. *C. songaricum* attracts *M. semenowi* and *M. desertora* to ingest or transport the seeds through the odor of the seeds and external elaiosome. The *M. semenowi* then defecate in their burrows, while the *M. desertora* transport the seeds to their nest.

In this study, volatile chemicals allowed the seeds of *C. songaricum* to be recognized by *M. semenowi* and *M. desertora* from a relatively long distance. When combined with non-volatile chemicals, this creates a more reliable and effective seed dispersal mechanism. As demonstrated in Fig 4, seeds have been recovered from the feces of *M. semenowi*, and *M. desertora* is also believed to be involved in transporting the seeds of *C. songaricum*. In this mutualistic, multi-species seed dispersal interaction, associative learning between odor and food reward mediates flexible communication between animals and plants [50,51].

In the current study, it remains to be determined whether the joint seed dispersal by *M. semenowi*, and *M. desertora* represents a unique case or is broadly applicable to the seed dispersal of other parasitic plants in similar habitats. This study proposes a model for the sequential and synergistic functions of multiple seed dispersal agents in the seed dispersal of parasitic plants, in which invertebrates, where invertebrates, previously overlooked, may play an important role. This is the first discovery of crosstalk among different species, which collaboratively facilitates the seed dispersal of *C. songaricum*.

This study sheds light on the role of *M. semenowi*, and *M. desertora* in the seed dispersal of *C. songaricum*, there are several limitations that warrant further investigation. First, the final destination of the seeds, including whether they are consumed, discarded, or dispersed by other agents such as lizards, remains unclear. Future studies should examine the fate of seeds beyond the *M. desertora* nests to clarify the broader dispersal mechanisms. Second, while *M. desertora* may consume seeds during food-scarce periods, the effect of this consumption on seed viability is yet to be determined. Additional research is needed to evaluate how *M. desertora* feeding impacts seed germination and establishment, particularly under different environmental conditions. Furthermore, the potential role of predators, such as lizards, in secondary or tertiary seed dispersal has not been explored, but their interactions with *M. desertora* could influence seed dispersal outcomes. Finally, the microclimatic conditions within *M. desertora* nests, such as temperature, humidity, and exposure to microbes, may affect seed viability, but this aspect was not addressed in the present study. Future research should investigate these factors to better understand their role in seed fate and plant establishment. These areas of exploration will provide a more comprehensive understanding of the ecological processes shaping seed dispersal in desert ecosystems.

## Conclusion

This study provides compelling evidence that *M. semenowi* and *M. desertora* play a significant role as seed dispersers for the parasitic plant *C. songaricum*. Through their foraging behaviors, beetles and ants facilitate the movement of seeds away from the parent plant, promoting

genetic diversity and reducing intraspecific competition. Our findings highlight the multi-modal nature of *C. songaricum* seed dispersal mechanism. *M. semenowi* directly consumes and transports seeds, while *M. desertora* acts as a secondary disperser by carrying seeds to their nests, thereby enhancing seed dispersal efficiency. The study also emphasizes the importance of olfactory cues in seed detection by *M. semenowi* and *M. desertora*, which are attracted to the distinctive odor emitted by the seeds of *C. songaricum*. These interactions exemplify the complex, mutualistic relationships between parasitic plants and their seed dispersers, providing new insights into seed dispersal mechanisms in desert ecosystems. Additionally, this research underscores the potential role of invertebrates, traditionally overlooked as seed dispersers, in maintaining the ecological balance of desert habitats. Further investigations are needed to explore the broader implications of these findings, particularly the role of secondary dispersal agents like lizards and the impact of environmental factors on seed viability. Overall, this study contributes to the growing body of knowledge on seed dispersal ecology and the adaptive strategies of parasitic plants, opening avenues for future research on the conservation and management of *C. songaricum* populations.

## Supporting information

**S1 Fig. (a-c) shows the visit frequency of *M. semenowi* and *M. desertora* at three study sites.** (a) Jilantai Town; (b) Yingen Sumu; (c) Ejina Banner. The study of three research sites found that beetles mainly nibbled on Suoyang in the morning (06:30-11:30) and afternoon (15:30-19:00). During the day (12:00-15:30), activity decreases sharply and the beetles disappear into the sand. Ants mainly consume seeds in the morning (06:30-12:30) and afternoon (15:00-19:30). During the day (12:30-15:00), activity decreases sharply, diving into the nest (Fig 3 and Fig 4). The difference of visiting time between ants and beetles in different research sites is related to the temperature of the research sites.
(TIF)

**S2 Fig. *C. songaricum* seeds in *M. semenowi* feces** (a) *M. semenowi* feces; (b) *C. songaricum* seeds; The area circled in red is *C. songaricum* seeds. (a), Bar = 2 mm; (b), Bar = 1 mm.
(TIF)

**S3 Fig. Microscopic observation of *C. songaricum* seeds.** (a) Natural seeds; (b) non-elaiosome seeds; (c) elaiosome comes into contact with water and wraps *C. songaricum* seeds; (d) Embryo. (a), (b), (c), Bars = 1 mm. (d), Bar = 500 μm. The elaiosome of *C. songaricum* seeds. When fruits are brown and dried, the dry pulp seems to play the role of an elaiosome.
(TIF)

**S4 Fig. C. songaricum seed viability in M. semenowi feces** (a-b) Seed embryos in feces after 12 hours; (c-d) Seed embryos in feces after 24 hours; (e-f) Seed embryos in feces after 48 hours; (a, c, e) Unstained seed embryos; (b, d, f) Stained seed embryos. (a), (c), (e), Bars = 2 mm. (b), (d), (f), Bars = 1 mm.
(TIF)

**S5 Fig. Study site natural *C. songaricum* seed vitality.** (Y) Yingen Sumu; (E) Ejina Banner; (J) Jilantai Town. The results of the seed vigor test at the three research sites showed that under natural conditions, the vigor of *C. songaricum* seeds at the three research sites was above 75%, including Yingen Sumu (84.5 ± 3.60%), Jilantai Town (81.1 ± 3.07%), and Ejin Banner (78.8 ± 3.46%).
(TIF)

**S1 Table. Study sites.**
(DOCX)

**S2 Table. The time, place and number of samples observed by visitors to the natural population of *C. songaricum*.**
(DOCX)

**S3 Table. The number of seeds that an *M. desertora* transports from the fleshy stem of *C. songaricum* back to the nest within a day.**
(DOCX)

**S4 Table. The average time it takes for an *M. desertora* to bite off a seed from the fleshy stem of *C. songaricum* and carry it back to the nest.**
(DOCX)

**S5 Table. The average time it takes for a *M. semenowi* to bite off seeds from the fleshy stem of *C. songaricum*.**
(DOCX)

**S6 Table. Scanning electron microscope observation of seeds in *M. semenowi* feces.**
(DOCX)

**S7 Table. Anatomical observation of seeds in the *M. semenowi* digestive tract.**
(DOCX)

**S8 Table. Visiting frequency of *M. semenowi* and *M. desertora* to the *C. songaricum* infructescence under different handles.**
(DOCX)

## Author contributions

**Conceptualization:** Zhi Wang, Dehui Wang, Guilin Chen.

**Data curation:** Zhi Wang, Huan Guan, Bingzhen Li, Kexin He.

**Formal analysis:** Zhi Wang, Guilin Chen.

**Investigation:** Zhi Wang, Kexin He, Zikang Jin, Guilin Chen.

**Methodology:** Zhi Wang, Huan Guan, Dehui Wang.

**Project administration:** Guilin Chen.

**Writing – original draft:** Zhi Wang.

**Writing – review & editing:** Huan Guan, Qianqian Zhang, Qing Chen, Guilin Chen.

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
