## [Decision Letter · Decision Letter 0]

6 Dec 2024

PONE-D-24-28571Endozoochory by the cooperation between beetles and ants in the holoparasitic plant Cynomorium songaricum in the deserts of Northwest ChinaPLOS ONE

Dear Dr. Chen,

Thank you for submitting your manuscript to PLOS ONE. After careful consideration, we feel that it has merit but does not fully meet PLOS ONE’s publication criteria as it currently stands. Therefore, we invite you to submit a revised version of the manuscript that addresses the points raised during the review process.

We look forward to receiving your revised manuscript.

Kind regards,

Rachid Bouharroud

Academic Editor

PLOS ONE

Journal Requirements:

4. We note that your Data Availability Statement is currently as follows: “All relevant data are within the manuscript and in Supporting Information files.”

Please confirm at this time whether or not your submission contains all raw data required to replicate the results of your study. Authors must share the “minimal data set” for their submission. PLOS defines the minimal data set to consist of the data required to replicate all study findings reported in the article, as well as related metadata and methods (https://journals.plos.org/plosone/s/data-availability#loc-minimal-data-set-definition). For example, authors should submit the following data: - The values behind the means, standard deviations and other measures reported; - The values used to build graphs; - The points extracted from images for analysis. Authors do not need to submit their entire data set if only a portion of the data was used in the reported study. If your submission does not contain these data, please either upload them as Supporting Information files or deposit them to a stable, public repository and provide us with the relevant URLs, DOIs, or accession numbers. For a list of recommended repositories, please see https://journals.plos.org/plosone/s/recommended-repositories. If there are ethical or legal restrictions on sharing a de-identified data set, please explain them in detail (e.g., data contain potentially sensitive information, data are owned by a third-party organization, etc.) and who has imposed them (e.g., an ethics committee). Please also provide contact information for a data access committee, ethics committee, or other institutional body to which data requests may be sent. If data are owned by a third party, please indicate how others may request data access.

5. We note that  Supporting Information Figure 1 in your submission contain map/satellite images which may be copyrighted. All PLOS content is published under the Creative Commons Attribution License (CC BY 4.0), which means that the manuscript, images, and Supporting Information files will be freely available online, and any third party is permitted to access, download, copy, distribute, and use these materials in any way, even commercially, with proper attribution. For these reasons, we cannot publish previously copyrighted maps or satellite images created using proprietary data, such as Google software (Google Maps, Street View, and Earth). For more information, see our copyright guidelines: http://journals.plos.org/plosone/s/licenses-and-copyright. We require you to either (a) present written permission from the copyright holder to publish these figures specifically under the CC BY 4.0 license, or (b) remove the figures from your submission:

a. You may seek permission from the original copyright holder of Supporting Information Figure 1 to publish the content specifically under the CC BY 4.0 license. We recommend that you contact the original copyright holder with the Content Permission Form (http://journals.plos.org/plosone/s/file?id=7c09/content-permission-form.pdf) and the following text: “I request permission for the open-access journal PLOS ONE to publish XXX under the Creative Commons Attribution License (CCAL) CC BY 4.0 (http://creativecommons.org/licenses/by/4.0/). Please be aware that this license allows unrestricted use and distribution, even commercially, by third parties. Please reply and provide explicit written permission to publish XXX under a CC BY license and complete the attached form.” Please upload the completed Content Permission Form or other proof of granted permissions as an "Other" file with your submission. In the figure caption of the copyrighted figure, please include the following text: “Reprinted from [ref] under a CC BY license, with permission from [name of publisher], original copyright [original copyright year].”

b. If you are unable to obtain permission from the original copyright holder to publish these figures under the CC BY 4.0 license or if the copyright holder’s requirements are incompatible with the CC BY 4.0 license, please either i) remove the figure or ii) supply a replacement figure that complies with the CC BY 4.0 license. Please check copyright information on all replacement figures and update the figure caption with source information. If applicable, please specify in the figure caption text when a figure is similar but not identical to the original image and is therefore for illustrative purposes only. The following resources for replacing copyrighted map figures may be helpful: USGS National Map Viewer (public domain): http://viewer.nationalmap.gov/viewer/ The Gateway to Astronaut Photography of Earth (public domain): http://eol.jsc.nasa.gov/sseop/clickmap/ Maps at the CIA (public domain): https://www.cia.gov/library/publications/the-world-factbook/index.html and https://www.cia.gov/library/publications/cia-maps-publications/index.html NASA Earth Observatory (public domain): http://earthobservatory.nasa.gov/ Landsat: http://landsat.visibleearth.nasa.gov/ USGS EROS (Earth Resources Observatory and Science (EROS) Center) (public domain): http://eros.usgs.gov/# Natural Earth (public domain): http://www.naturalearthdata.com/

Additional Editor Comments:

Dear Author

As suggested by reviewers, the manuscript deserve to be published in PlosOne but need some improvements. Please give special attention to temperature conditions, behaviour of Messor genus and lizard interactions.

Good luck

Reviewers' comments:

Reviewer's Responses to Questions

**Comments to the Author**

1. Is the manuscript technically sound, and do the data support the conclusions?

Reviewer #1: Yes

Reviewer #2: Partly

2. Has the statistical analysis been performed appropriately and rigorously? 

Reviewer #1: No

Reviewer #2: Yes

3. Have the authors made all data underlying the findings in their manuscript fully available?

Reviewer #1: Yes

Reviewer #2: Yes

4. Is the manuscript presented in an intelligible fashion and written in standard English?

Reviewer #1: Yes

Reviewer #2: No

5. Review Comments to the Author

Reviewer #1: The manuscript needs, in my opinion, minor corrections.

Here are some critical points that need to be addressed in the revised version of the manuscript:

Introduction: Authors need to insert more recent information (articles published) on the subjects covered.

Material and Methods: Author should clear all the necessary queries and changes recommended (especially experiments design and statistical analysis) because experiment design is too important in research.

Results: Some paragraphs in the results section need to be rewritten or clarified for better comprehension.

Conclusion: add conclusion section with the mains finding.

All specifics comment are grouped in the table below

Reviewer #2: The study is highly relevant; I would just recommend reviewing the following comments:

A. An in-depth linguistic revision is indispensable. While I have corrected some errors, many others still need to be addressed to enhance the overall linguistic quality of the manuscript.

B. It is essential to determine the scientific name of the beetle mentioned in the study, given the importance of this information to the research.

C. No information is provided regarding the name "Messor desetus" in the document (see the comment in the PDF). This point absolutely needs to be reviewed. In the absence of the author's name and the year of description, it is unclear which species is being referred to. This name does not appear in Barry Bolton's online ant catalog. You are likely referring to Messor desertor He & Song, 2009, which is a valid species found in your study area.

D. The protocol for ant collection and their transfer to artificial nests must be detailed, including air or surface soil temperatures.

E. Temperature data are missing in the study, although this factor generally determines insect activity, especially for Messor and beetles. Discussing foraging periods without mentioning temperatures is highly relative. The ant activity periods presented in the results should include the corresponding soil temperatures (measured at 1 cm depth) or, alternatively, air temperatures.

F. Upon reading the results on ants, it appears that they feed on the infructescences and seeds of *C. songaricum* outside the nest, particularly when you mention that midday heat forces them to return to their nests. If this is the case, it would be preferable to state it explicitly. For the genus Messor, the consumption of seeds outside the nest is very rare. Their behavior primarily focuses on transporting and storing seeds for internal processing and consumption. It is therefore crucial to verify this information.

G. The role of ants from the genus Messor in seed management requires an in-depth analysis of their ecological impacts. Unlike beetles, it is essential to specify the destination of the seeds collected by these ants. Lizards, acting as tertiary dispersal agents (Fig. 6), can consume Messor ants, raising the question of the impact of this interaction on seed germination, particularly if the seeds are ingested by the ants themselves—something that remains uncertain. Additionally, the seeds collected might be consumed by the ants (by grinding them), especially during periods of food scarcity, or discarded with waste outside the nest, where they could be dispersed by other means or germinate in situ. Finally, it would be relevant to investigate their ability to germinate in abandoned ant nests or in failed colonies.

For the minor comments in the PDF, I point out:

L14: (C. songaricum) Not necessary, however, mention the author and the year of description.

L25: No information about the name "Messor desetus" was provided in the document. In the absence of the author's name and the year of description, it is unclear which species you are referring to. This name does not appear in *An Online Catalog of the Ants of the World* by Barry Bolton. You are probably referring to *Messor desertor* He & Song, 2009, which is a valid species present in your study area.

L58: "10 familiesé. You have listed only 8 families in parentheses; kindly double-check this information.

Why not add the Cytinaceae and Orobanchaceae families to the list?

L60 : add "and"

L65 : Are you referring to Suetsugu (2018b)? If not, please specify the year of publication.

L72: genus Cynomorium from, and not C. Cynomorium.

L111: in 2021 and 2022.

L113: Cite the reference for these data.

L:113-114: It is recommended to use the Köppen-Geiger classification (Kottek et al., 2006).

L141: (were counted (n=178, 51.29%).)This is a result and should be moved to the results section.

L150: Can you justify this duration? 48h

L160-161: This section requires greater detail. How many foragers did you collect? How did you ensure they belonged to the same colony (to avoid conflicts)? Were they collected directly from the plant or during their return to the nest? Was there a foraging column present? How was the artificial nest designed? Are there any photos available (as supplementary material)?

L161-163: We fed them 100 C. songaricum seeds. After 6 h, 12 h, 24 h, and 48 h of observations, we collected the C. songaricum seeds from the ant nests and counted the number of intact seeds (n=300).

L167: due to chewing processes

L170: We then used

L173: De Vega and De Oliveira

L179: greasy smell?

L182: The infructescence was covered

L185: was cavered

L192: from 6:00 am to 8:00 pm

L196: They visit the plant, not the seeds!

L198: Under what weather conditions? Specifically, at what temperature? Temperature is the primary factor determining the foraging activity of ants, not the availability of seeds.

L201-203: Here too, it is essential to have data on nocturnal temperatures for the same reason mentioned above.

L205: situation ????

L206: study

Table1: Coleptera, not in italics. Myrmicinae, not in italics, desertus in italics (after cheking the name). No information about the name "Messor desetus" was provided in the document. In the absence of the author's name and the year of description, it is unclear which species you are referring to. This name does not appear in *An Online Catalog of the Ants of the World* by Barry Bolton. You are probably referring to *Messor desertor* He & Song, 2009, which is a valid species present in your study area.

L210: Close the parenthesis.

L211: deleta Celsius.

L211: beetle activity decreases

L2016: feeds on

L229: of C. songaricum

L230: the analysis

L236: 48 h after the beetles had eaten

L237-238: There was no significant difference between the two feeding modes.

L288: (a-c) Anatomy

L290: (j-l) Anatomy

L294-295: This requires specifying the temperature ranges that correspond to this activity.

L198-299: When reading this text, it appears that the ants feed on the infructescences of seeds or the infructescences of *C. songaricum* outside the nest, especially when you mention that the midday heat forces them to return to their nests. If this is the case, it would be better to state it explicitly, as it is well known for the genus *Messor* that the consumption of seeds outside the nest is very rare. Their behavior is primarily focused on transporting and storing seeds for internal processing and consumption. It is therefore crucial to verify this information.

L306-310: This should be mentioned in theMaterials and Methods section.

L312: and ants consumed only the style attached to the seeds.

Table3: Seed type

L373: The role of ants from the genus Messor in seed management requires an in-depth analysis of their ecological impacts. Unlike beetles, it is essential to specify the destination of the seeds collected by these ants. Lizards, acting as tertiary dispersal agents (Fig. 6), can consume *Messor* ants, raising the question of the impact of this interaction on seed germination, particularly if the seeds are ingested by the ants themselves—something that remains uncertain. Additionally, the seeds collected might be consumed by the ants (by grinding them), especially during periods of food scarcity, or discarded with waste outside the nest, where they could be dispersed by other means or germinate in situ. Finally, it would be relevant to investigate their ability to germinate in abandoned ant nests or in failed colonies.

Moreover, the microclimatic conditions in the storage chambers of the nests, such as temperature, humidity, and exposure to microbial agents, may also influence seed viability. These interactions could promote germination by altering the seed coat or, conversely, accelerate seed deterioration. These complex processes, involving seed manipulation, environmental conditions, and ant behavior, deserve further study to better understand the ecological role of Messor ants as seed dispersers and managers.

L399:"to eat", As already mentioned: collecting seeds is one thing, but feeding on them on-site is another.

L401: Idem

L444-445: Here, you are referring to transportation, not on-site feeding. Please refer to the previous comments for clarification.

L544: De Vega

L577: De Vega

L587: De Castro-Arrazola

6. PLOS authors have the option to publish the peer review history of their article (what does this mean? ). If published, this will include your full peer review and any attached files.

**Do you want your identity to be public for this peer review?** For information about this choice, including consent withdrawal, please see our Privacy Policy .

Reviewer #1: **Yes: ** Abdelhadi AJERRAR

Reviewer #2: No

---

## [Author Response · Author response to Decision Letter 0]

22 Jan 2025

Dear editor and reviewers,

Thank you for offering us an opportunity to improve the quality of our submitted manuscript entitled “Endozoochory by the cooperation between beetles and ants in the holoparasitic plant Cynomorium songaricum in the deserts of Northwest China” (PONE-D-24-28571). We sincerely thank the reviewer for the valuable comments and constructive suggestions, which have significantly contributed to improving the quality of our manuscript. We appreciate the reviewer’s time and effort in carefully evaluating our work.

In response to the comments provided, we have carefully revised the manuscript. Below is a summary of the changes made to address the main concerns:

Responses to Reviewer #1：

Introduction: Authors need to insert more recent information (articles published) on the subjects covered.

Response: Thank you for your valuable comments. In response to the reviewer's suggestions, we have updated the introduction section by incorporating recent literature to ensure the discussion reflects the latest advancements in the relevant research areas.

Materials and Methods: Author should clear all the necessary queries and changes recommended (especially experiments design and statistical analysis) because experiment design is too important in research.

Response: We gratefully appreciate your valuable comment. We have clarified the experimental design and statistical analysis methods to enhance transparency and reproducibility. Additional details have been included to address potential queries.

Results: Some paragraphs in the results section need to be rewritten or clarified for better comprehension.

Response: Some paragraphs in the Results section of the revised manuscript, such as lines 469 to 513 on pages 22-24 and lines 610 to 614 on page 29 have been rewritten and clarified to improve readability and ensure accurate interpretation of the findings.

Conclusion: Add conclusion section with the mains finding.

Response: A dedicated conclusion section has been added to summarize the main findings and their implications.

We have provided detailed responses to each specific comment in the following sections, and all corresponding revisions have been highlighted in the manuscript for ease of review. We hope that the revised version meets the reviewer’s expectations. Once again, we deeply appreciate the reviewer’s thoughtful suggestions and valuable feedback.

Comment 1:

Line 38: Please delete Kingdom

Response: Thank you for pointing this out. We have deleted the term "Kingdom" as suggested. The change has been made on page 3, line 51 in the revised manuscript.

Comment 2:

Line 42: Parasitical plants are divided into hemiparasite and holoparasite plants according to their light and ability. Please add more details

Response: Thank you for your insightful comment. In response, we have expanded the section to provide more details on hemiparasitic and holoparasitic plants, including examples and their physiological characteristics. The revised text now reads as follows:

“Hemiparasites, such as Viscum L. (mistletoe) and Striga Lour. (witchweed), possess chlorophyll and are capable of limited photosynthesis, yet they still depend on their host plants for water and nutrients. In contrast, holoparasites, including Orobanche L. (broomrape) and Cuscuta L. (dodder), lack chlorophyll entirely and rely exclusively on their host plants for both carbon and mineral nutrients.”

This addition has been included on pages 3, lines 57 to 62 in the revised manuscript.

Comment 3:

Line 72-72: Please add citation to this sentence.

Response: Thank you for your suggestion. We have added the appropriate citation to support the statement as requested.

The reference is as follows:

Chen Guilin. Chinese Cynomorium songaricum. 1st ed. Hohhot: Science Press, 2016.

These citations have been added on page 5, lines 97 in the revised manuscript [14].

Comment 4:

Line 80: Please put proper word instead management.

Response: Thank you for pointing this out. We have replaced the word "management" with more appropriate terminology to enhance clarity and precision. We have changed “These components are known to offer benefits such as antioxidant and antiviral properties, besides helping in the management of obesity, diabetes, Alzheimer's disease, and enhancing memory function” to “These components are known to possess antioxidant and antiviral properties, as well as to aid in the prevention and treatment of obesity, diabetes, Alzheimer's disease, and enhancing memory function”.

This change has been made on page 5, lines 102-105 in the revised manuscript.

Comment 5:

Line 87: Delete “Our”

Response: Thank you for your suggestion. We have removed the term "Our" from the manuscript as requested.

This change has been made on page 6, line 112 in the revised manuscript.

Comment 6:

Line 114: Please add the name of three sites were research was conducted with a brief description of each one; Please add a map showing the three sites.

Response: Thank you for your suggestion. In response, we have added the names and descriptions of the three research sites, including their coordinates and brief ecological characteristics as follows:

“Study site 1: The research base in Jilan Tai Town, Alxa Left Banner, located in the southeastern Tengger Desert (N39°36', E105°05'), with an elevation of 1200-1400 m. This site is characterized by arid conditions and a diverse plant community, with C. songaricum occurring naturally and parasitizing Nitraria L. as its host; Study Site 2: The research base in Yinggen Sumu, Alxa Left Banner, located in the northeastern Tengger Desert (N38°27', E105°23'), with an elevation of 1200-1400 m. Similar to site 1, this region features semi-arid conditions and supports wild populations of C. songaricum, which parasitize Nitraria L.; Study Site 3: The research base in Ejina Banner, Alxa League, situated in the northwestern Badain Jaran Desert, with an elevation ranging from 1200 to 1700 m. This site is characterized by harsh desert conditions and sparse vegetation, with C. songaricum growing as a wild species and parasitizing Nitraria L.”

S1 Fig shows the locations of these three research sites. However, due to map protection, the editor has suggested that we delete the map. As a result, this part of the content cannot be presented as an image. We have provided the research site information in the form of an attached table (S1 Table).

These changes have been incorporated on pages 7-8, lines 148 to 160 in the revised manuscript.

Comment 7:

Line 115: What do you mean by Sutra?

Response: Thank you for pointing this out. To avoid any confusion, we have deleted the term "Sutra" from the manuscript.

This change has been made on page 7, line 148 in the revised manuscript.

Comment 8:

Line 117-120: The C. songaricum population is a wild population that relies on Nitrariaceae as its host plant. The C. songaricum seeds were in contact with or closely located in the ground, suggesting that they could be consumed by animals that live near to the ground” please remove this sentence from Materials and Methods to Introduction.

Response: Thank you for your suggestion. We have moved the sentences as requested, transferring them from the Materials and Methods section to the Introduction for better contextual alignment. We have inserted this sentence into page 6 lines 115 to 117 of the Introduction.

Comment 9:

Line 121: Please update recent citation. Why did you choose remote camera for investigation? I think investigation took a long time? Can you explain why

Response: Thank you for your insightful comments and questions. We have updated the citation with more recent references, as requested. These updates have been incorporated on page 9, line 184 in the revised manuscript.

References:

Monteza-Moreno CM, Rodriguez-Castro L, Castillo-Caballero PL, Toribio E,Saltonstall K. Arboreal camera trapping sheds light on seed dispersal of the world’s only epiphytic gymnosperm: Zamia pseudoparasitica. Ecology and Evolution. 2022; 12: e8769. https://doi.org/10.1002/ece3.8769

Campos CM, Velez S, Miguel MF, Papú S, Cona MI. Studying the quantity component of seed dispersal effectiveness from exclosure treatments and camera trapping. Ecology and Evolution. 2018; 8: 5470-5479. https://doi.org/10.1002/ece3.4068

Regarding the use of remote cameras, we chose them to monitor animal activity, particularly whether animals like mice, rabbits, and yellow sheep were gnawing on C. songaricum. Infrared cameras also helped observe animal visits at night. No visits were observed, which minimized field disturbance.

The extended investigation duration can be explained by:

Remote Locations: The study sites in the Tengger and Badain Jaran Deserts are difficult to access.

Weather Conditions: Extreme temperatures and seasonal rainfall limited fieldwork.

Data Collection: Long-term monitoring and sampling for phenological studies were time-consuming.

Ecological Factors: Complex interactions between C. songaricum and its host plant, Nitraria, required detailed, seasonal observations.

Resource Constraints: Limited resources also contributed to the longer duration.

These factors explain why the study took longer than expected.

Comment 10:

Line 128: Please add measures under bars for photos.

Response: Thank you for your suggestion. We have added the requested measures under the bars for the photos as per your recommendation. The revised figures now include the appropriate scale bars for better clarity and understanding of the data presented.

Comment 11:

Line 131: Throughout, use the species name of studied insects instead of beetles and ants.

Response: Thank you for pointing this out. We have replaced the generic terms "beetles" and "ants" with the specific species names throughout the manuscript. The revised text now refers to Mantichorula semenowi Reitter, 1888 (beetles) and Messor desertora He & Song, 2009 (ants).

Comment 12:

Line 133: To avoid repetition put “interactions between C. songaricum and the two studied insects”. Instead of “, interactions between C. songaricum and beetles, and C. songaricum and ants”

Response: Thank you for your suggestion. We have revised the sentence to avoid repetition, as recommended. We have modified the sentence accordingly from “interactions between C. songaricum and beetles, and C. songaricum and ants were assessed by their activity times and behaviors” to “interactions between C. songaricum and the two studied insects were assessed based on their activity duration and behaviors”.

This change has been made on page 10, line 220 in the revised manuscript.

Comment 13:

Line 135-141: Please add more details concerning beetles foraging

Experiment design, indicate the following information:

The number of visits you did in 2021 and 2022

The frequency of visits (weekly, Monthly...etc)

Did you visit the three sites during one day or each day you visit one site.

Duration of each visit to catch coleoptera. Did you use a standard duration?

Were beetles captured equally in the three sites? indicate the differences if any

Response: Thank you for your suggestions. Here's a more concise and direct response to your comment:

From 2021 to 2022, we conducted systematic observations at three research sites from June to August each year. Each site was observed three times per year, totaling nine observations annually, with one observation per month on average. Each site observation lasted 2-3 days, and subsequent site visits occurred only after completing observations at the previous site. The observation period for beetles and ants was consistent, spanning from sunrise to sunset for comprehensive data collection. Beetle captures were consistent across the three sites, with no significant differences observed. These details have been added to the Materials and Methods section (S2 Table).

This change has been made on pages 10-12, lines 211 to 247 in the revised manuscript.

Comment 14:

Line 146-147: Please mention the number of repetition to measure this duration.

How did you calculate the number of seeds carried out the anthill?

Response: Thank you for your comment. Here's the revised response with added clarity and conciseness:

Throughout the experiment, observations were conducted continuously from sunrise to sunset over three consecutive days at each research station, repeated three times across the study period, totaling nine days of observation. Two researchers tracked the movement of seeds by ants, noting every ant that began nibbling on seeds and eventually transporting them to the nest. The time it took for each ant to carry one seed from the C. songaricum flowers to the nest was recorded as a valid observation.

For calculating the number of seeds transported, random sampling was used. Each day, 10 ants were randomly selected from the ant groups visiting the C. songaricum plants. The carrying process of each ant was followed from the point of seed handling to entry into the nest, with the time recorded. The average time for ants to carry seeds to the nest was then calculated. By factoring in the overall frequency of ant visits and the total duration of their activity, the number of seeds transported to the nest was determined.

Specific details can be found in the Materials and Methods and Results section (S3 Table). This change has been made lines 236 to 247 on pages 11-12 and lines 442 to 445 on page 21 in the revised manuscript.

Comment 15:

Line 135-147: Please remove all results from this section.

Response: Thank you for your suggestion. We have removed all results in this section as requested. The relevant results have been supplemented in the Results section.

This change has been made on page 20, lines 437 to 445 in the revised manuscript.

Comment 16:

Line 152-153: Please rephrase the sentence: 30 beetles captured in the wild were placed in the same breeding room in groups of 10, for a total of three breeding rooms (n=30).

Response: Thank you for your suggestion. We have rephrased the sentence for clarity as follows:

Mixed rearing: Thirty M. semenowi captured from the wild were placed in three independent rearing rooms, with 10 M. semenowi in each room (The rearing room is a transparent cuboid, measuring 20 cm in length, 10 cm in width and 15 cm in height. Its top is fitted with a grid cover, facilitating gas exchange and observation).

This change has been made on page 13, lines 284 to 287 in the revised manuscript.

Comment 17:

Line 148-165: Please mention breeding parameters in laboratory for both ants and coleoptera.

Response: Thank you for your suggestion. We have added the following information regarding the breeding parameters in the laboratory:

"Both M. semenowi and M. aciculatus were subjected to feeding experiments in a laboratory setting under controlled conditions. Specifically, for M. semenowi, the breeding chamber was maintained at a constant temperature of 25 ± 2°C, with relative humidity ranging from 40% to 60%. For M. aciculatus, the artificial nests were kept under the same temperature conditions, with humidity precisely controlled between 50% and 60%. During the experiment, both species had unrestricted access to water, and C. songaricum seeds were the sole food source."

This additional information has been included in the Materials and Methods section for clarity. This change has been made on page 13, line 267 to 273 in the revised manuscript.

Comment 18:

Line 181-192: Please mention the number of repetitions adopted in experiment of odor and view signals.

Response: Thank you for your insightful comment. To address this, we have provided the details regarding the number of repetitions for the odor and view signal experiments in the revised manuscript.

The experiment was conducted with two treatment groups, each repeated for 7 consecutive days, as follows:

“Two treatment groups were set up in the experiment, each repeated for 7 consecutive days. The number and types of visitors were recorded daily from sunrise to sunset. Specifically, we tested the visitors' response to the fruit sequence (view signals) and seed scent (odor signals). Two treatment groups were set up in the wild. Treatment A involved covering the infructescence with yellow gauze mesh bags to isolate the visual signal (color) of

---

## [Editor Report · Decision Letter 1]

28 Jan 2025

Endozoochory by the cooperation between beetles and ants in the holoparasitic plant Cynomorium songaricum in the deserts of Northwest China

PONE-D-24-28571R1

Dear Dr. Chen,

We’re pleased to inform you that your manuscript has been judged scientifically suitable for publication and will be formally accepted for publication once it meets all outstanding technical requirements.

Kind regards,

Rachid Bouharroud

Academic Editor

PLOS ONE
---

## [Editor Report · Acceptance letter]

PONE-D-24-28571R1

PLOS ONE

Dear Dr. Chen,

I'm pleased to inform you that your manuscript has been deemed suitable for publication in PLOS ONE. Congratulations! Your manuscript is now being handed over to our production team.

Kind regards,

on behalf of

Dr. Rachid Bouharroud

Academic Editor

PLOS ONE